# Modified in-vitro AATCC-100 procedure to measure viable bacteria from wound dressings

**Sang Hyuk Lee**[1,2,3], **Thomas Glover**[1,2,3], **Nathan Lavey**[2,3], **Xiao Fu**[2,3], **Marc Donohue**[1], **Enusha Karunasena**[2]*

**1** Department of Chemical and Biomolecular Engineering, Whiting School of Engineering, Johns Hopkins University, Baltimore, MD, United States of America, **2** Division of Biology, Chemistry, and Materials Science, Office of Science and Engineering Laboratories, Center for Devices and Radiological Health, US Food and Drug Administration (FDA), Silver Spring, MD, United States of America, **3** US Department of Energy, Oak Ridge Institute for Science and Education, Office of Science, Oak Ridge, TN, United States of America

* enusha.karunasena@fda.hhs.gov, enusha.karunasena@gmail.com

**Data Availability Statement:** All relevant data are within the manuscript and its Supporting Information files.

**Funding:** This project and financial support was provided by the Critical Path program at US Food

## Abstract

Chronic wounds are reoccurring healthcare problems in the United States and cost up to $50 billion annually. Improper wound care results in complications such as wound debridement, surgical amputation, and increased morbidity/ mortality due to opportunistic infections. To eliminate wound infections, many antimicrobial dressings are developed and submitted to FDA for evaluation. AATCC-100 is a standard method widely used to evaluate cloth wound dressings. This method, requires enrichment, followed by culturing to measure the concentration of culturable organisms; a caveat to this method could result in neglected viable but nonculturable (VBNC) bacteria and overestimate the antimicrobial properties of wound dressings. Therefore, the objectives of this study were to assess this accepted protocol with quantitative real-time polymerase chain reaction (qRT-PCR), to measure time dependent antimicrobial efficacy of wound dressing, and to examine for potential viable bacteria but non-culturable as compared with traditional plating methods. The test organisms included opportunistic pathogens: *Pseudomonas aeruginosa* (ATCC 15692) and *Staphylococcus aureus* (ATCC 43300). To mimic a wound dressing environment, samples of commercially available wound dressings (McKesson Inc.) with silver ion (positive control) and dressings without silver ion (positive control) were assessed under sterile conditions. All samples were examined by the original protocol (the extended AATCC-100 method) and qRT-PCR. The expression of specific housekeeping genes was measured (*proC* for *P. aeruginosa* and *16s rRNA* for *S. aureus*). Based on these tests, log reduction of experimental conditions was compared to identify time dependent and precise antimicrobial properties from wound dressing samples. These results showed antimicrobial properties of wound dressings diminished as incubation days are increased for both methods from day 1 PCR result of 4.31 ± 0.54 and day 1 plating result of 6.31 ± 3.04 to day 3 PCR result of 1.22 ± 0.97 and day 3 plating result of 5.89 ± 2.41. These results show that data from qRT-PCR generally produced lower standard deviation than that of culture methods, hence shown to be

and Drug Administration / Center for Device and Radiological Health. The funders had no role in study design, data collection and analysis, decision to publish, or preparation of the manuscript.

**Competing interests:** The authors have declared that no competing interests exist.

more precise. Complementary parallel analysis of samples using both methods better characterized antimicrobial properties of the tested samples.

## Introduction

An increase in the use of antimicrobial dressings to combat opportunistic wound infections, accounts for at least $28 B USD [1] in annual healthcare costs. Wound dressings have been used for a millennia to treat bacterial infections. Opportunistic wound infections are caused by commensal organisms that may become invasive in a vulnerable host, such as immunocompromised patients undergoing surgery. These pathogens may localize to the site of injury (surgical site infections), invade vital organs (resulting in pneumonia/endocarditis) or enter the bloodstream (causing bacteremia) [2] which can result in fatal complications. Although antibiotic and biocide treatments may be effective in the short term, over time bacterial resistance inevitably limits the efficacy of these treatments. Therefore, standardized protocols that can accurately assess the time-dependent antimicrobial properties of wound dressings are critical to patient care.

Today, antimicrobial dressings are used in clinical settings and homecare. These dressings are typically conjugated with agents to enhance biocidal activity. Conjugation is achieved via polymerization, nanofabrication, or ionization. In the case of ionization, the ionized biocidal agent, such as silver, is exchanged for anions that form the textile of the dressing fabric. In this investigation, calcium alginate dressings are used, therefore the anions in the dressings are [3] mostly from alginate. When the dressing is immersed in the wound exudate, the silver ions are released into the wound exudate via ionic exchange with cations in the exudate, [4] such as sodium. This interaction of the wound with the ions and the dressing results in fluid uptake, which may also be driven by protein chelation and surface detachment due to degradation of the fibers. These silver ions bind to bacterial membranes and induce cell death in a variety of ways. It should be noted that in addition to the silver ions, the dressing fibers have an antimicrobial effect. Specifically, as the dressing absorbs wound fluid and become hemostatic, bacteria are entrapped between the fibers [4] and become more susceptible to the biocide.

There are quantitative and qualitative methods to evaluate wound dressing antimicrobial efficacy. Unlike qualitative methods, quantitative methods traditionally rely on cell count by culturing bacterial cells harvested from wound dressings in a harvesting media. A common quantitative method determines biocidal efficacy by measuring the area of the zone of inhibition (ZOI), in which no bacterial growth is observable. This is done by swabbing the harvesting media or agar plate with bacterial culture, in which the wound dressing is placed either on the surface of the media or within the media. A smaller zone of inhibition is indicative of reduced [5] biocidal efficacy.

The AATCC-100 is the most common qualitative method. The method includes exponential phase cell growth, inoculation onto wound dressing samples, neutralization of the biocidal agent, and harvesting of incubated cells with a harvesting media. Cell concentration is determined by counting the colony forming units (CFUs) in a cell culture assay using agar plates, and log reduction by normalization to the control dressing [6] and/or to the starting inoculate.

The two organisms recommended for testing in AATCC-100 are *Pseudomonas aeruginosa* and *Staphylococcus aureus*—opportunistic pathogens commonly associated with wound infections. *P. aeruginosa*, is a gram negative, rod shaped bacterium with flagella assisted motility, and is the second most common cause of pneumonia [7] from nosocomial settings. *P. aeruginosa* has flagellum assisted mobility and secretes pyocyanin and pyoverdine, which are green

and blue fluorescent toxins that are closely related to virulence [8] and proliferation. They are able to secrete large amounts of exopolysaccharide chain and DNA in the formation of biofilms [9] as cells proliferate. A biofilm is a self-constructed three-dimensional community of bacteria that enables bacteria to live on a variety of substrates [10] and protects a community. *P. aeruginosa* uses quorum sensing to monitor and respond to their own population density, which is essential for adapting to environmental conditions [11] such as oxidative stress and nutrient deprivation. Secretion of toxins, biofilm generation, and quorum sensing systems can extend persistence of cells under biocidal stress. Therefore, using *P. aeruginosa* as a model pathogen and understanding its adaptability to antimicrobials aids in the development of a more reliable evaluation protocol.

A similarly prevalent wound pathogen is *S. aureus*, a nonmotile gram positive bacteria of circular shape, recognized for its characteristic gold pigmentation, due to its secretion of Staphyloxanthin. *S. aureus* is the leading cause of surgical site infections (SSIs) and pneumonia, and the second leading [12] cause of blood stream infections. Since the 1990s, cases of community-acquired methicillin-resistant *S. aureus* (CA-MRSA) have increased [12–14], suggesting that the threat of infection is no longer isolated to nosocomial settings. Like *P. aeruginosa*, *S. aureus* is highly sensitive to spatial and temporal signaling, and it engages in quorum sensing [15] via the accessory gene regulator (agr). Profiling of *S. aureus* virulence factors, such as spA and leukotoxin-F demonstrates strong sensitivity to physiological signaling, such as pH, salt concentration, oxidative stress, ion concentration, and the presence of immune cells. When exposed to biocidal pressure, it may be that virulence is turned off to extend survivability until more favorable growth conditions arise. The viable-but-non-culturable (VBNC) phenotype is well documented, and in response to unfavorable environmental conditions, [16–21] such as antibiotic pressure, resulting in persistence.

VBNC, a metabolically dormant state in which bacteria are alive but fail to grow on standard laboratory media [22], is another reason AATCC– 100 may be incomplete as a test standard. VBNC is known as a survival status in response to adverse environmental conditions such as lack of oxygen or nutrients, presence of antibiotics or antimicrobial agents, and unfavorable pH or temperature. VBNC cells may be induced by random fluctuations in gene expression [23] like those of persister cells. Environmental stress can also induce persister and VBNC cells. However, unlike persister cells, VBNC is temporary and reversible [22] under appropriate conditions. The reversal process of VBNC is also known as resuscitation and this process is crucial towards understanding bacterial behavior in the presence of environmental stress such as silver ion. Accordingly, methods that depend on culture-based assays alone may underestimate both quantity and adaptability of the infective agent.

Metal ion containing wound dressings are over-the-counter, commercially available products for wound infections. Companies may submit these products to the US Food and Drug Administration (FDA) for evaluation regarding their safety and efficacy, for FDA clearance. To ensure that these products meet proposed indication of use, standard protocols are used for product testing. Currently, one of the most commonly used methods is AATCC-100: Antimicrobial Fabric Test (AATCC-100) which specifically serves as a protocol to evaluate the antimicrobial properties [6] of textiles. However, the original protocol, developed in the 1960s, has limitations which may result in less effective product performance testing.

Both qualitative and quantitative methods have limitations. They can underestimate bacterial quantity and misidentify bacterial species, leading to inaccurate conclusions regarding product efficacy of the wound dressings investigated. Although quantitative methods are usually preferred, the traditional approach is inherently retrospective, as it relies on assay results with a 24-hour incubation period. Due to product duration testing for 24 hours, the method also limits monitoring of bacterial persistence. Lastly, these classical methods do not capture

phenotypes that may be induced by wound dressing exposure which may inform more effective treatment regimens. For these reasons, alternative methods to culturing, like qRT-PCR, may offer some improvements to the evaluation of antimicrobial dressings. When fully optimized, the assay can be sensitive to both species and phenotypes [24,25] that are overlooked by traditional quantitative methods.

To this end, our objective was to assess this accepted protocol with quantitative real-time polymerase chain reaction (qRT-PCR), to measure time dependent antimicrobial efficacy of wound dressings, and to examine for potential viable bacteria but non-culturable bacteria as compared with traditional plating methods. This investigation features an extended version of this method, with 3- and 7-day incubation with addition of simulated wound fluids. This study describes a protocol that could expand upon AATCC-100, featuring multiple time points for testing that are representative of the product's 'Indications of Use'. This latter assay can track changes in antimicrobial properties of dressings over a longer duration of incubation.

## Materials and methods

### Extended AATCC-100 protocol

The methods of extended AATCC– 100 protocols are adapted from the original AATCC– 100 methods [26] and outlined in the following sections.

http://dx.doi.org/10.17504/protocols.io.rm7vzxo5xgx1/v1 [PROTOCOL DOI]

**Dressing preparation.**   McKesson Calcium Alginate Dressings® (McKesson Inc) were used to prepare samples. The experimental group consisted of dressing with silver ions and the control group were dressing without antimicrobial silver ions. Each dressing sheet was cut into four one square inch sections, one section was allocated for sterile control and the remaining three for biological replicates. This process was repeated for each incubation period amounting to 24 pieces in total (8 samples each, were tested for incubation periods of 1, 3, and 7 days respectively).

Each dressing was added to a sterile, 8 oz Nalgene™ bottle (Nalgene Inc, 24 bottles in total). The Nalgene bottle was sterilized and free of moisture, prior to use. Each dressing was hydrated with 1.2 mL of simulated wound fluid (SWF), a medium commonly incorporated into dressing test methods to simulate an in vivo wound environment. SWF contained equal volumes of Bolton broth (BB; Thermo Fisher Scientific Inc) and fetal bovine serum (FBS; Thermo Fisher Scientific Inc). Bottles were capped and covered with aluminum foil to minimize moisture loss.

**Bacterial culture and sample preparation.**   An overnight culture of *S. aureus* ATCC-43300 was introduced into tryptic soy broth and grown to exponential phase ($1 \times 10^8$ CFU / mL) and measured by optical density at wavelength 600 nm ($OD_{600}$; 0.8). Aliquots of 250 μL of culture was added to each biological sample. Following inoculation, each sample was incubated at 37˚C for 1, 3, or 7 days. Similarly, the exponential phase of *P. aeruginosa* ATCC 15692 was prepared and inoculated to separate sets of wound dressing samples with $OD_{600}$ of around 0.1 ($1 \times 10^8$ CFU / mL).

**Harvesting cells from wound dressings.**   Following 1, 3, and 7 days of incubation, 25 mL of D/E neutralization buffer was added to each of the eight samples for a given time point. Each jar was vortexed for 90 seconds. Aliquots of 15 mL from each sample were flash-frozen in liquid nitrogen and stored at -80˚C. For each sample, the remaining extract was serially diluted and plated. All dilutions from biological replicates and controls were plated in triplicate.

### Dressing hydration

Following 4 days of incubation for samples inoculated with *S. aureus*, day 7 dressings were rehydrated with 500 μL SWF. It was determined observationally that day 1 and 3 dressings

remained sufficiently fluid for the duration of their incubation and did not need to be rehydrated. The rehydration of day 7 jars was to ensure consistent hydration of the dressings that is representative of a wound environment. For samples inoculated with *P. aeruginosa*, day 3 samples were hydrated at day 2; day 7 samples were hydrated at day 2 and day 5.

**Log reduction.** For the quantification of antimicrobial properties, log reduction was determined for each extraction day and for both experimental and control groups. The equation for log reduction is the following:

Log Reduction = $Log_{10}$ (Concentration of Starting Inoculate)–$Log_{10}$ (Concentration of Harvested Bacteria)

**Comparison of log reduction.** To examine the effect of silver ion compared to controls (samples without silver ion), comparison of log reductions were calculated with the following equation:

Comparison of Log Reduction = $Log_{10}$ (Average Concentration of Silver Ion Present Samples)–$Log_{10}$ (Average Concentration of Silver Ion Absent Samples)

**RNA extraction.** Approximately, 5 mL overnight cultures of *S. aureus* (ATCC-43300 ($OD_{600}$~1.2)) and *P. aeruginosa* (ATCC 15692 ($OD_{600}$~0.12) were centrifuged. The pellets were resuspended in an equal volume of RNA protection reagent (RNAprotect Bacteria Reagent®, Qiagen Inc). Seven μL of β-mercaptoethanol (BME) was added and centrifugation and resuspension were repeated 2 to 3 times. The resuspension medium was a mixture of proteinase K and DEPC water. BME and proteinase K were added to degrade any RNases and proteins in the sample. Following the final centrifugation, the pellet was resuspended in 700 μL buffer RLT (Buffer RLT®, Qiagen Inc), a lysis buffer. Five hundred microliters of 100% ethanol was added to the suspension.

**RNA purification.** Three 500 μL aliquots of a suspension were added into sample tubes. RNA extraction was performed using an RNAeasy kit® (Qiagen Inc) and the purity and yield was confirmed spectrophotometrically (Nanodrop One™, Thermo Fisher Scientific Inc). RNA was cleaned and concentrated using Zymo's Clean & Concentrate Kit® (Zymo Research Inc).

**cDNA synthesis.** cDNA was synthesized using Superscript III Kit® (Invitrogen Inc) on a Thermocycler™ (Applied Biosystems Inc). Cycling conditions were the following: 10 minutes at 25˚C followed by 50 minutes at 50˚C and reaction termination at 85˚C for 5 minutes. This was followed by RNase digestion at 20 minutes for 37˚C. A no reverse transcriptase control was included to account for DNA contamination. Random hexamer primers were used to unselectively reverse transcribe present mRNA sequences. The purity and yield of the cDNA was confirmed spectrophotometrically, using Nanodrop.

**qRT-PCR.** Amplicons (RxnReady® Primer Pools, Integrated DNA Technologies Inc) specific to *P. aeruginosa* or *S. aureus* at a concentration of 500 nM along with 10 μL of SYBR® green master mix (iTaq Universal STBR Green Supermix®, Biorad Inc) were assayed with cDNA at a concentration of approximately 500 ng / μL. Amplicons for *P. aeruginosa* with annealing temperature of 55˚C were F' `ACCCCGCATAGCGTTCATC` and R' `GGAGACGAT CAGTTGCTCCG`; for *S. aureus* with annealing temperature of 53˚C and forward primer was F' `CCATGAAGTCGGAATCGCTAG` and reverse R' `ACTCCCATGGTGTGACGG`.

Assays were performed on a Biorad CFX Connect 96 Real-Time PCR Detection System™ (Bio-Rad Inc) and reaction conditions were the following: heat-activation for 50˚C for 10 minutes, 5 minutes at 95˚C followed by 50 cycles of denaturation at 95˚C for 5 seconds and annealing/extension at 53˚C for 30 seconds. This protocol was repeated at three different annealing temperatures, using the 'BioRad CFX Connect 96' temperature gradient protocol. Reaction efficiencies were determined by evaluating the slopes of the calibration curve. The non-template control (NTC) threshold cycle (Ct) for both primers were compared to determine the annealing temperature and primer pairs to optimize reaction efficiencies, establish

quantification limits, or lower limit of quantification (LLOQ) for the assay. Ct for each dilution were compared with cell concentration to generate a standard curve for quantification. This process was repeated for extracts from wound dressings of known cell concentration, ensuring the assay could be performed to measure bacterial cells extracted from wound dressings. The LLOQ measure depended on the NTC since the NTC signal was an off-target amplification signal. Any Ct that is above the NTC Ct is taken to be indeterminate.

**Housekeeping genes: proC and 16S rRNA.** The reference genes for absolute qRT-PCR analysis were proC and 16s rRNA (Integrated DNA Technologies Inc.; IDT). proC was selected as the target gene given its relevance to metabolic activities [27] by *P. aeruginosa* such as amino acid biosynthesis. For *S. aureus*, 16S rRNA was targeted for its stability [28] and longer half-life. These two genes were selected as housekeeping genes to then conduct expression analysis and develop a calibration curve (log CFU vs threshold cycle).

**Absolute quantification (qRT-PCR).** qRT-PCR results of cDNA analysis were used to generate a calibration curve. Non-template control (NTC) was included to account for background fluorescence. These calibration curves were used to extrapolate the log (CFU) concentration for viable bacterial cells. This methodology allowed for log and comparison of log reduction analysis between cDNA samples from bacteria inoculated onto wound dressings.

**Statistical analysis.** Data are expressed as mean ± standard error of the mean (SEM) of triplicates. All statistical analyses were performed using R version 4.1.0 at a significance level of 0.05. Data visualization was performed with GraphPad Prism 9 (GraphPad Software Inc., USA). Two-way ANOVA was used to analyze the effect of the two quantitative methods (qRT-PCR and Cell Culture) and incubation day (Day 1, 3, and 7) on the antimicrobial properties of silver ion containing wound dressings. The Student's paired t-test was used to compare antimicrobial properties of factors including silver ion (with silver ion (positive control) and without silver ion (positive control)), quantitative method (qRT-PCR and Cell Culture), and incubation day (Day 1, 3, and 7). The P values were calculated by two-tailed paired t test with Bonferroni corrections between P. Significance codes: '****' P-value <0.0001, '***' P-value <0.001; '**' P-value <0.01; '*' P-value <0.05.

## Results

Bacteria Growth Characterization of *P. aeruginosa* (ATCC 15692)

Figs 1 and 2 showed CFU / mL and $OD_{600}$ readings, to measure *P. aeruginosa* (ATCC 15692) and demonstrated bacterial growth at lag, log, and stationary phases. Based on these growth curves, the inoculate bacterial population were harvested after 3 hours at log phase of growth.

Bacteria Growth Characterization of *S. aureus* (ATCC 43300)

Figs 3 and 4 showed CFU / mL and $OD_{600}$ readings, to measure growth of *S. aureus* (ATCC 43300) demonstrated bacterial growth at lag, log, and stationary phases. Based on these growth curves, the inoculate bacterial population were harvested after 3 hours at log phase of growth.

Calibration Curve of *P. aeruginosa* (ATCC 15692)

The calibration curves for *P. aeruginosa* (ATCC 15692) in Fig 5 showed high linear correlation between CFU / mL and $OD_{600}$. The target inoculation population was $1X10^7$ CFU / mL for original and extended AATCC-100 and the calibration curve generated were used to determine bacterial populations.

Calibration Curve of *S. aureus* (ATCC 43300)

The calibration curves for *S. aureus* (ATCC 43300) in Fig 6 showed high linear correlation between CFU / mL and $OD_{600}$. The target inoculation population was $1X10^7$ CFU / mL for original and extended AATCC-100 and the calibration curve generated was used to determine bacterial populations and then compared with cell culturability.

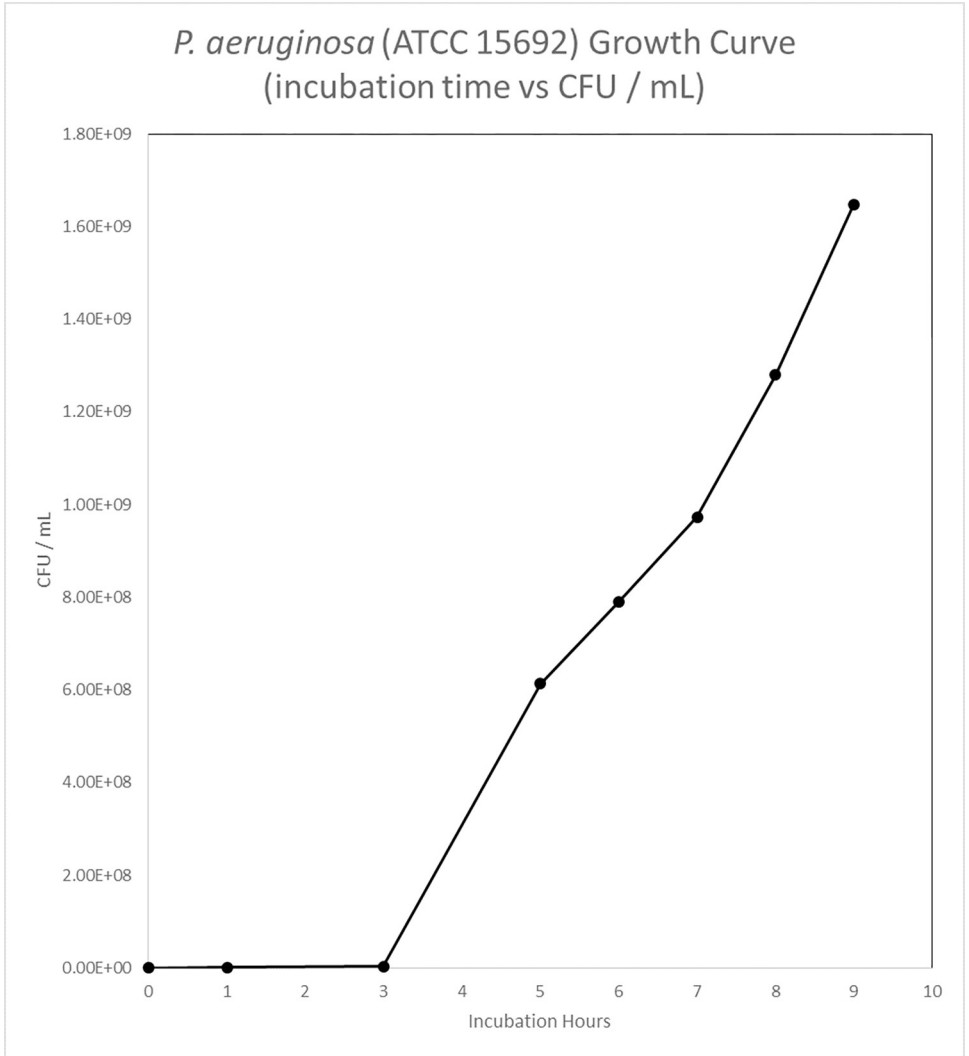

**Fig 1. P. aeruginosa (ATCC 15692) growth curve (incubation time (hr) vs CFU / mL).**

Fig 7 is the calibration curves of log (CFU / mL) vs threshold cycle for target housekeeping genes: proC and 16s rRNA. The linear fit equations and $R^2$ values are shown respectively. The proC calibration curve demonstrated the best fit $R^2$ value for *P. aeruginosa* while 16s rRNA demonstrated the highest $R^2$ value for *S. aureus*. $R^2$ values close to 1 shows direct linear correlation between log (CFU / mL) and threshold cycle values.

Fig 8 is a pairwise comparison of the expressions from P. aeruginosa (in logarithmic scale) was conducted. Expressions were measured using a total of four methods, categorized by the presence (Ag+) or absence (Ag-) of silver ions combined with two detection techniques: PCR and Cell Culturability. The specific methods include Ag+ PCR, Ag- PCR, Ag+ Cell Culturability, and Ag- Cell Culturability. Data was collected over three time points: day 1, day 3, and day 7. A two-way ANOVA, considering both '**quantitative method**' and 'day' as factors, was applied for analysis. The results indicated a significant influence of the '**quantitative method**' factor on expression (p < 0.001), while the 'day' factor was not statistically significant (p = 0.148). Post hoc pairwise analysis using Tukey's HSD test revealed significant differences in expressions among certain methods. The log reduction of Ag + PCR from Day 1, 3, and 7

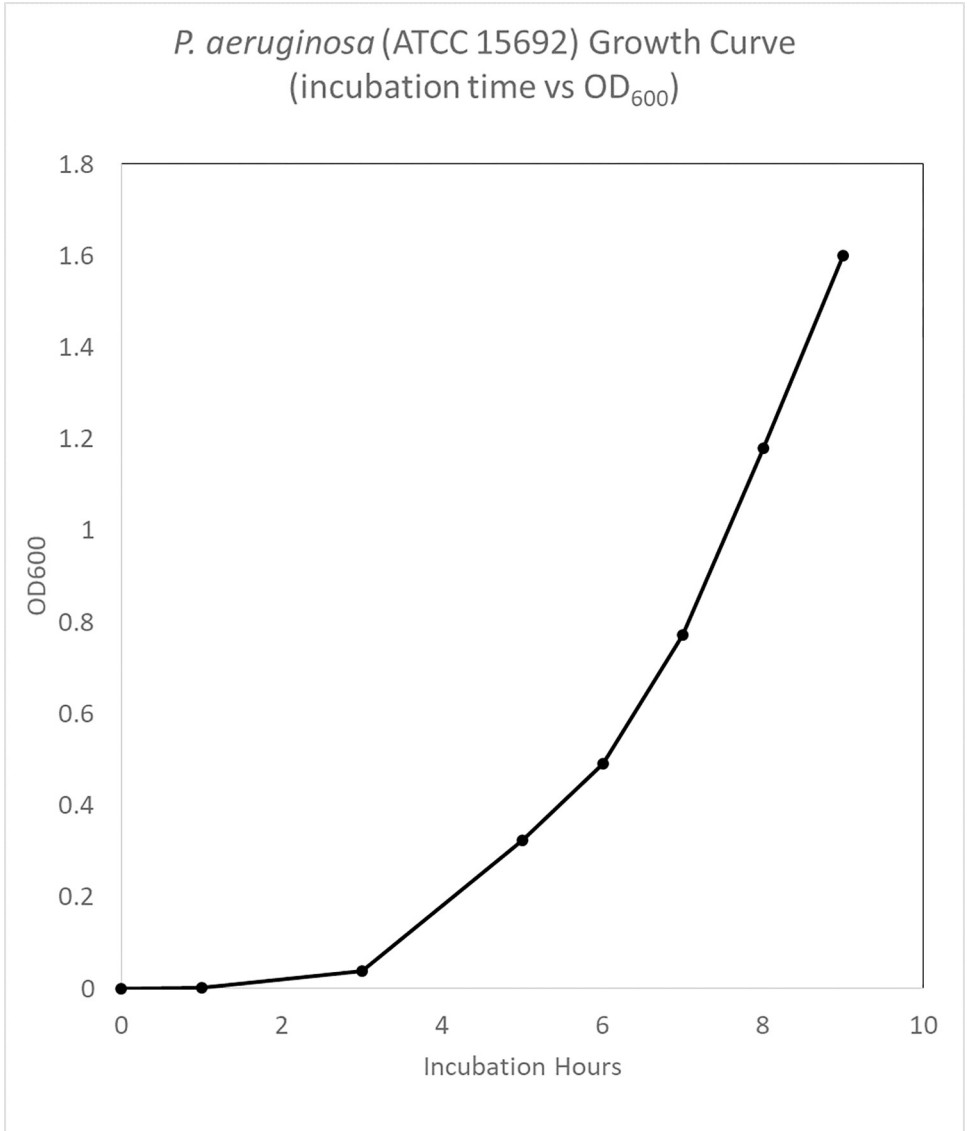

**Fig 2. P. aeruginosa (ATCC 15692) growth curve (incubation hours vs OD$_{600}$).**

were $3.27 \pm 0.36$, $0.75 \pm 0.54$, $2.18 \pm 2.09$, respectively. The log reduction of Ag—PCR from Day 1, 3, and 7 were $-1.04 \pm 0.18$, $-0.47 \pm 0.42$, $0.12 \pm 0.34$, respectively. The log reduction of Ag + Cell Culture from Day 1, 3, and 7 were $6.28 \pm 2.29$, $5.28 \pm 2.03$, $0.09 \pm 0.06$, respectively. The log reduction of Ag—Cell Culture from Day 1, 3, and 7 were $-0.03 \pm 0.75$, $-0.61 \pm 0.39$, $0.23 \pm 0.06$, respectively.

Fig 9 is a comparison of the expressions from *P. aeruginosa* (in logarithmic scale) using four distinct quantative methods, resulting from combinations of the presence (Ag+) or absence (Ag-) of silver ions and the detection techniques employed: PCR and Cell Culturability. Specifically, the quantitative methods are Ag+ PCR, Ag- PCR, Ag+ Cell Culturability, and Ag- Cell Culturability. A paired t-test was used to further elucidate these within-method changes across the time points.

Fig 10 is a comparison of changes in expressions from *P. aeruginosa* (in logarithmic scale). Expressions were measured using two specific methods that evaluated the difference in Log

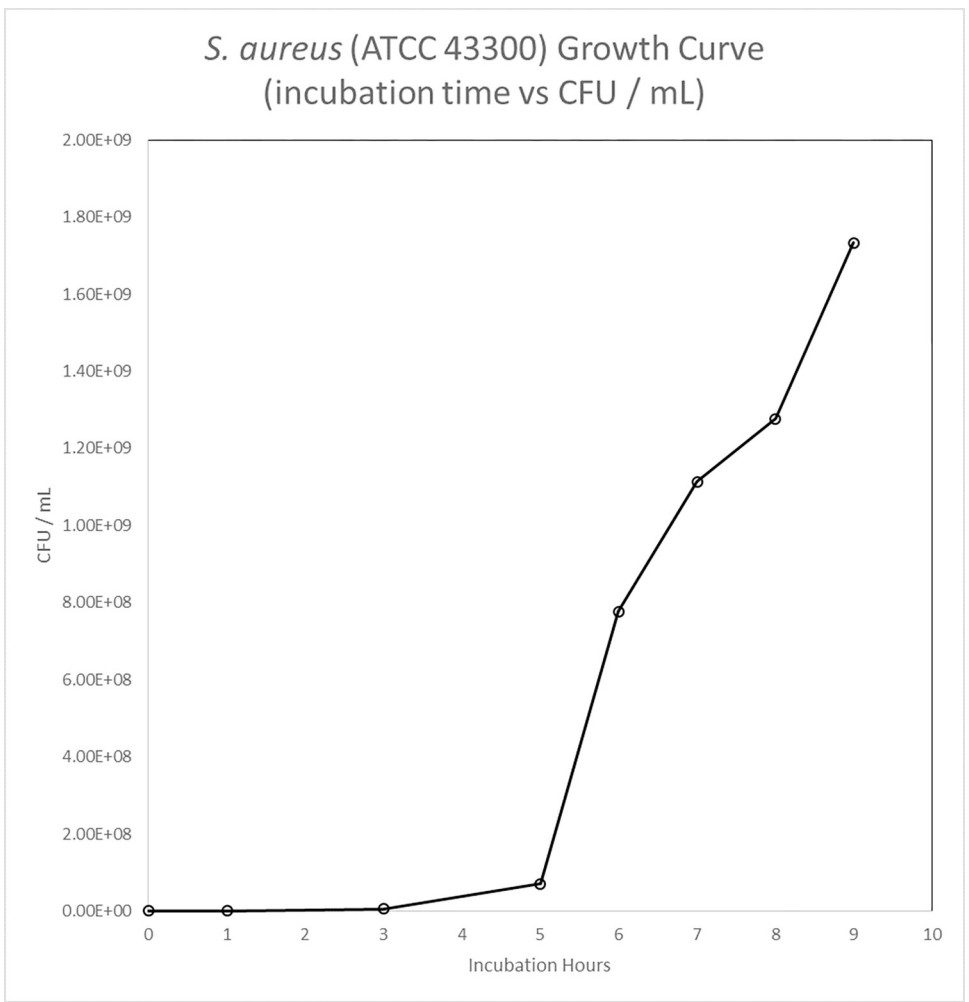

**Fig 3. S. aureus (ATCC 43300) growth curve (incubation time (hr) vs CFU / mL).**

Reduction within the presence (Ag+) or absence (Ag-) of silver ions: the quantitative methods of qRT-PCR and culture. Data was collected over three time points: day 1, day 3, and day 7. A two-way ANOVA, considering both 'quantitative method' and 'day' as factors, was applied for analysis. The results indicated a significant influence of the 'day' factor on expression ($p < 0.001$) and also a significant impact from the 'quantitative method' factor ($p = 0.00848$). Post hoc pairwise analysis using Tukey's HSD test was presented in the figure, revealing significant differences in expressions between the methods across the days. The comparison of log reduction of PCR from Day 1, 3, and 7 were 4.31 ± 0.54, 1.22 ± 0.97, 2.06 ± 2.43, respectively. The comparison of log reduction of bacterial cell culturability from Day 1, 3, and 7 were 6.31 ± 3.04, 5.89 ± 2.41, -0.15 ± 0.12, respectively.

Fig 11 is a comparison of changes in expressions from *P. aeruginosa* (in logarithmic scale). Expressions were measured using two specific methods that evaluated the difference in Log Reduction within the presence (Ag+) or absence (Ag-) of silver ions: the quantitative methods of qRT-PCR and culture. Data was collected over three time points: day 1, day 3, and day 7. While the data underwent the same two-way ANOVA analysis, considering both 'quantitative method' and 'day' as factors, the emphasis is on comparing changes in expressions over time

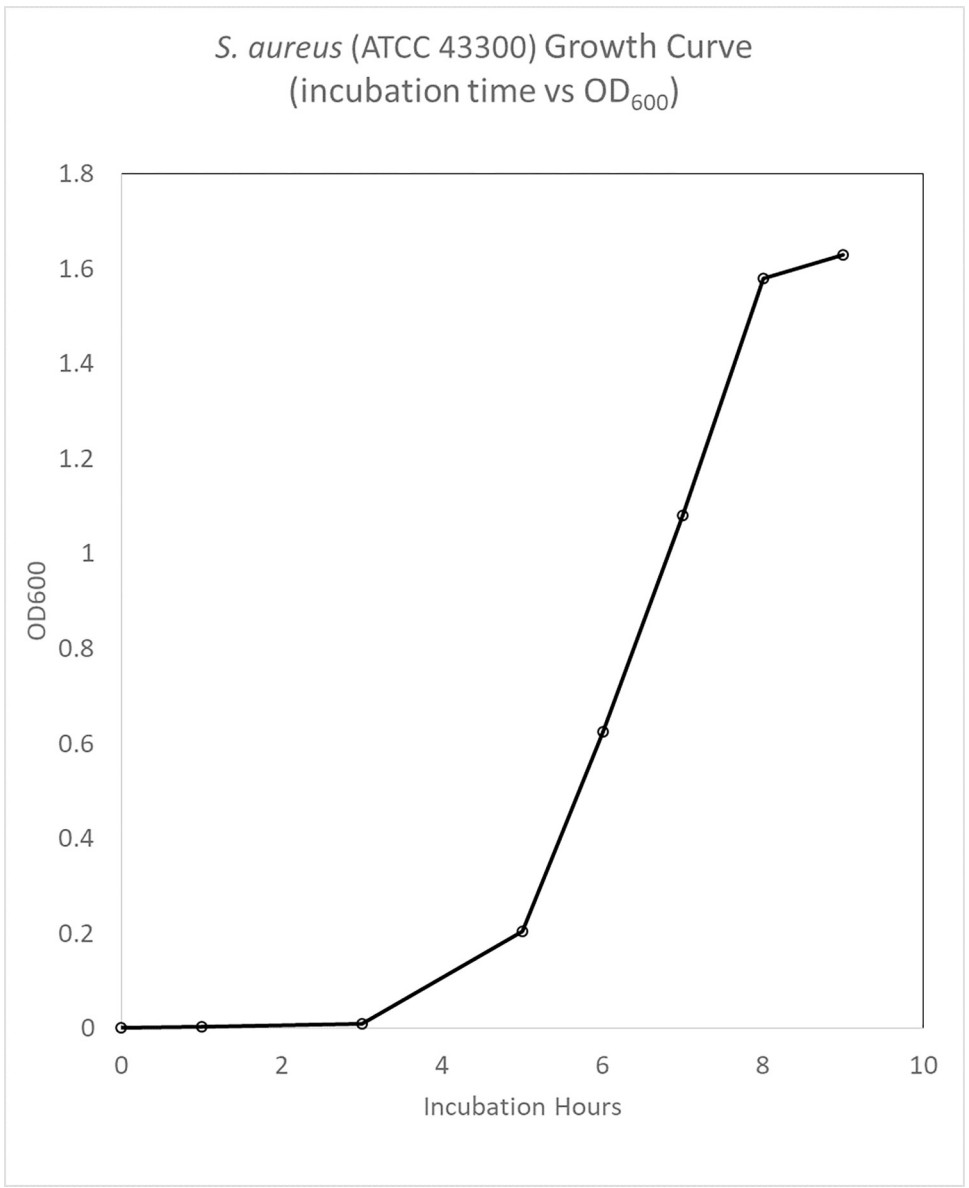

**Fig 4. S. aureus (ATCC 43300) growth curve (incubation time (hr) vs OD$_{600}$).**

within these different quantitative methods. A paired t-test was used to further elucidate these within-method changes across the time points.

Fig 12 is a pairwise comparison of expressions from *S. aureus* (in logarithmic scale). Expressions were measured using a total of four quantitative methods, categorized by the presence (Ag+) or absence (Ag-) of silver ions combined with two detection techniques: PCR and Cell Culturability. The specific methods include Ag+ PCR, Ag- PCR, Ag+ Cell Culturability, and Ag- Cell Culturability. Data was collected over three time points: day 1, day 3, and day 7. A two-way ANOVA, considering both 'quantitative method' and 'day' as factors, was applied for analysis. The results indicated a significant influence of the 'method' factor on expression (p < 0.001), while the 'day' factor was not statistically significant (p = 0.0948). Post hoc pairwise analysis using Tukey's HSD test revealed significant differences in expressions among the

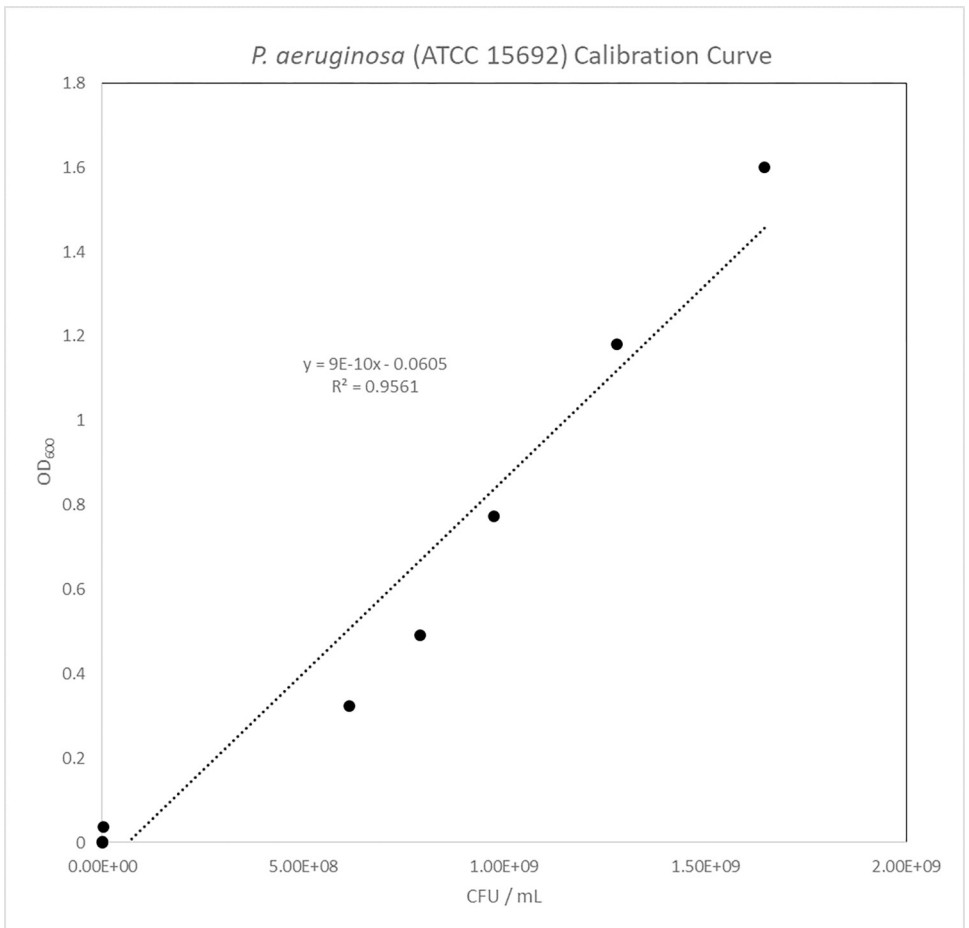

**Fig 5. P. aeruginosa (ATCC 15692) calibration curve.**

4 methods. The log reduction of Ag + PCR from Day 1, 3, and 7 were 4.13 ± 0.81, 1.76 ± 0.25, 3.52 ± 0.32, respectively. The log reduction of Ag—PCR from Day 1, 3, and 7 were 2.79 ± 1.18, 1.83 ± 0.19, 3.12 ± 1.64, respectively. The log reduction of Ag + Cell Culture from Day 1, 3, and 7 were 4.00 ± 0.38, 2.37 ± 1.51, 1.88 ± 1.67, respectively. The log reduction of Ag—Cell Culture from Day 1, 3, and 7 were -0.04 ± 0.24, 0.95 ± 0.27, 1.00 ± 0.28, respectively.

Fig 13 is a comparison of the expressions from *S. aureus* (in logarithmic scale) using four distinct methods, resulting from combinations of the presence (Ag+) or absence (Ag-) of silver ions and the detection techniques employed: PCR and Cell Culturability. Specifically, the quantitative methods are Ag+ PCR, Ag- PCR, Ag+ Cell Culturability, and Ag- Cell Culturability. While the data underwent the same two-way ANOVA analysis as, considering both 'quantitative method' and 'day' as factors, the emphasis is on comparing changes in expressions over time within these different quantitative methods.

Fig 14 is a comparison of changes in expressions from *S. aureus* (in logarithmic scale). Expressions were measured using two specific methods that evaluated the difference in Log Reduction within the presence (Ag+) or absence (Ag-) of silver ions: the quantitative methods of qRT-PCR and culture. Data was collected over three time points: day 1, day 3, and day 7. A two-way ANOVA, considering both 'quantitative method' and 'day' as factors, was applied for analysis. The results indicated a significant influence of the 'day' factor on expression ($p < 0.001$) and also a significant impact from the 'method' factor ($p < 0.001$). Post hoc

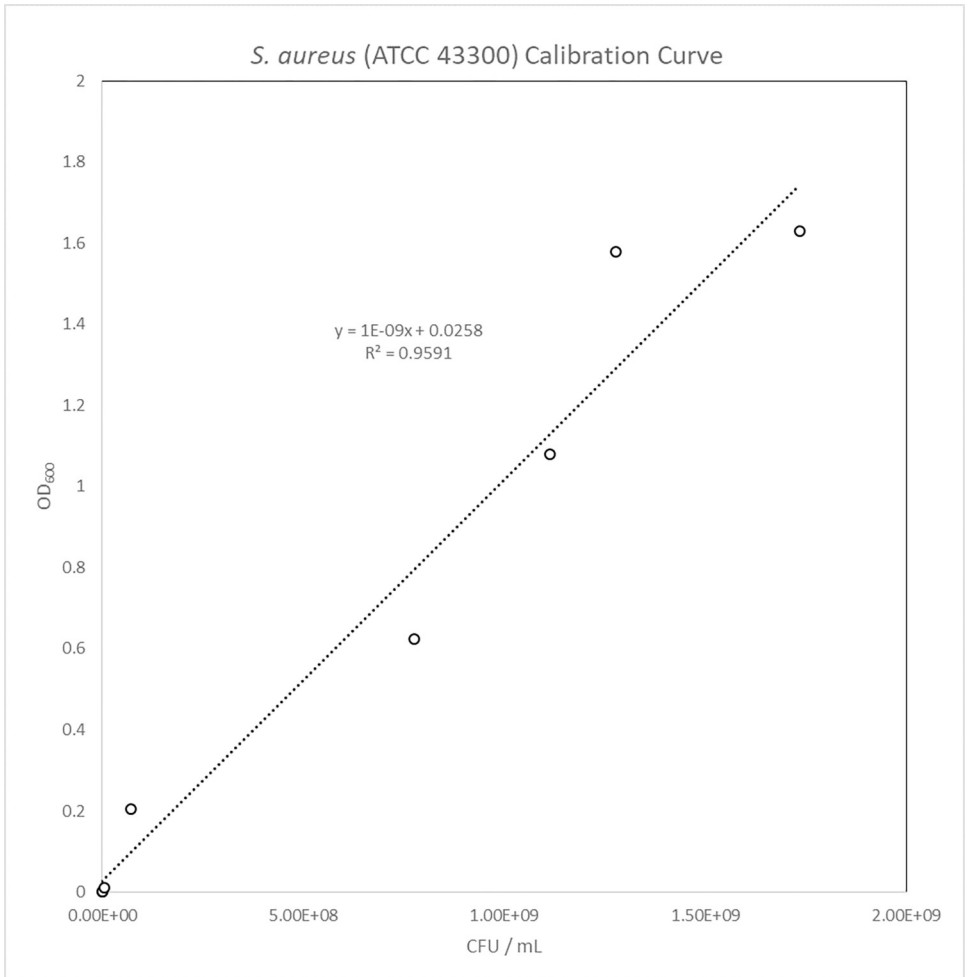

**Fig 6. S. aureus (ATCC 43300) calibration curve.**

pairwise analysis using Tukey's HSD test was presented in the figure, revealing significant differences in expressions between the quantitative methods across the days. The comparison of log reduction of PCR from Day 1, 3, and 7 were $1.35 \pm 1.99$, $-0.07 \pm 0.44$, $0.40 \pm 1.96$, respectively. The comparison of log reduction of bacterial cell culture from Day 1, 3, and 7 were $4.04 \pm 0.62$, $1.42 \pm 1.78$, $0.87 \pm 1.95$, respectively.

Fig 15 is a comparison of the changes in expressions from *S. aureus* (in logarithmic scale). Expressions were measured using two specific methods that evaluated the difference in Log Reduction within the presence (Ag+) or absence (Ag-) of silver ions: the quantitative methods of qRT-PCR and culture. Data was collected over three time points: day 1, day 3, and day 7. While the data underwent the same two-way ANOVA analysis, considering both 'method' and 'day' as factors, the emphasis is on comparing changes in expressions within each method at different time point. A paired t-test was used to further elucidate these within-method changes across the time points.

## Discussion

This investigation compared log and comparison of log reduction of qRT-PCR and bacterial culture assays for testing wound dressings for time-dependent antimicrobial efficacy. For both

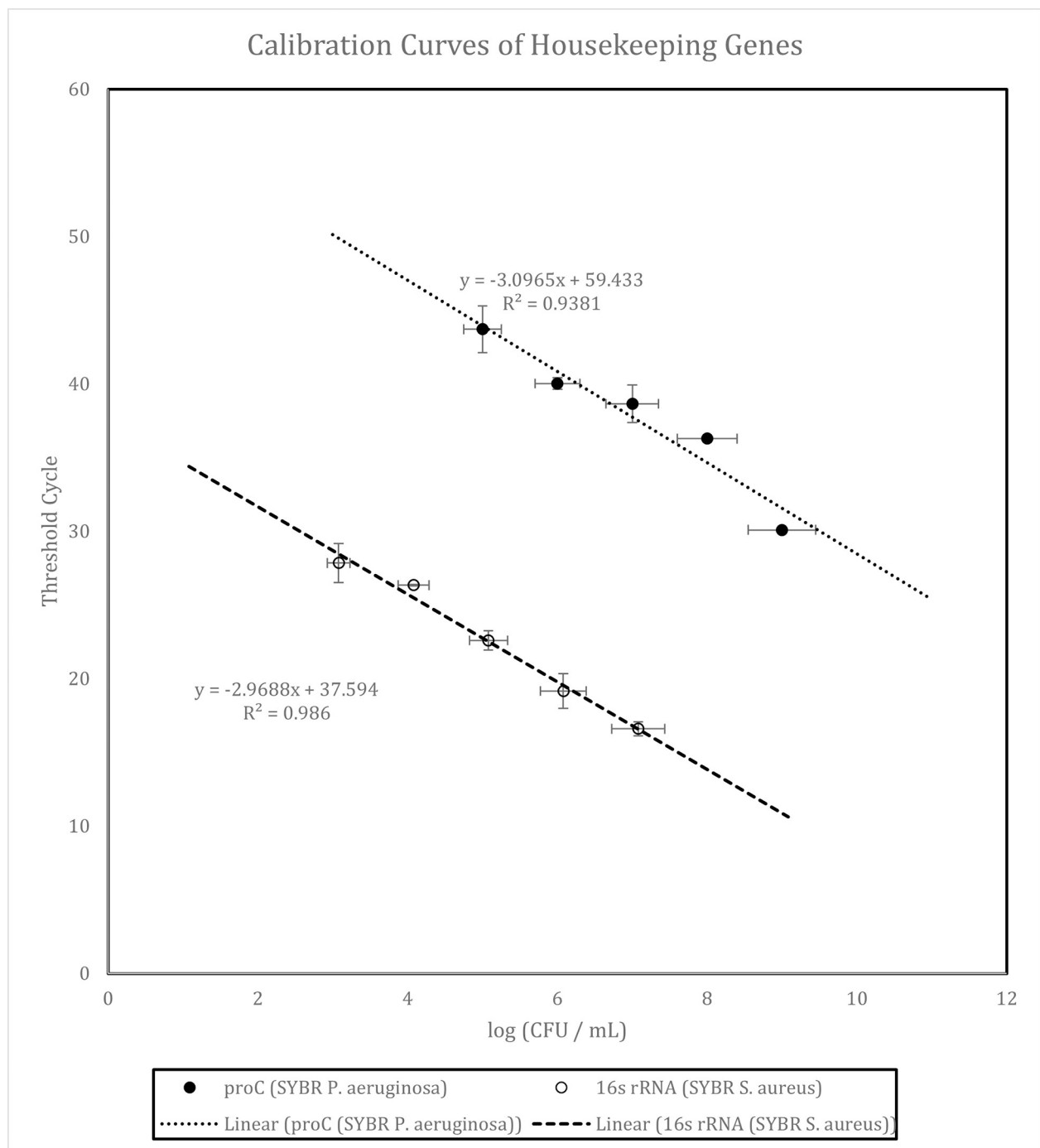

**Fig 7. Comparison of calibration curves of target genes: (proC; SYBR and 16s rRNA; SYBR).**

opportunistic pathogens tested, results suggested that silver ion containing wound dressings exhibited the highest antimicrobial effect after 24 hours, however the effect diminished by days 3 and 7. Similar to the original [26,29] and extended AATCC-100 protocol and the reliance on cell culture for microbial viability/culturability, qRT-PCR could be used as a quantitative method. Although, cell culture has advantages such as broad technique familiarity, cost, and

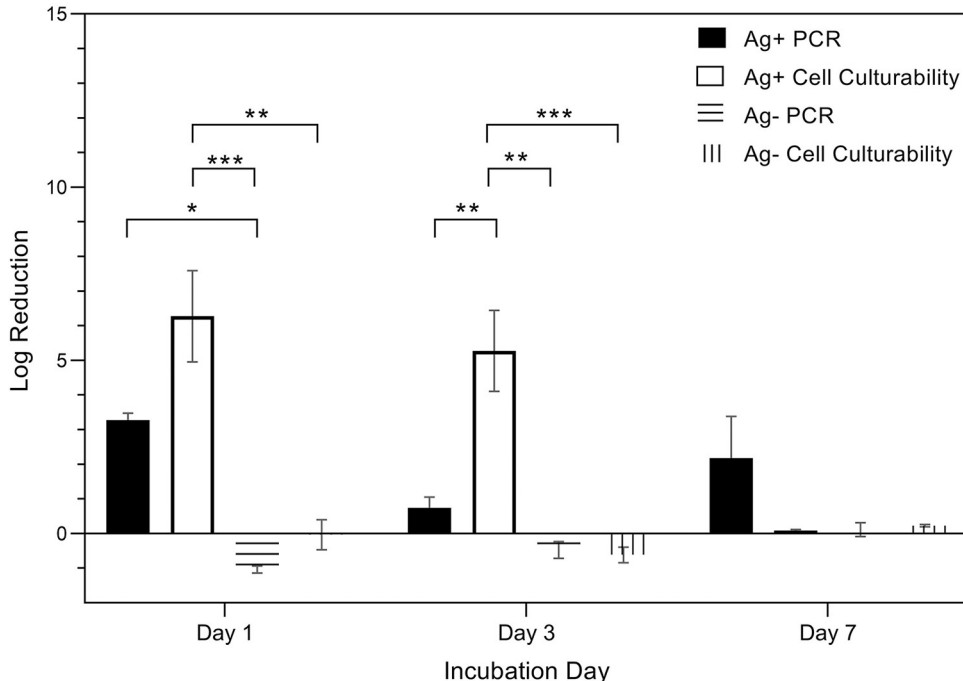

**Fig 8. Log reduction of P. aeruginosa by quantitative methods.**

accessibility—non-culturable but viable organisms were less likely to be captured through plating methods, compared with molecular biology assays.

The bacterial cell culture assay results indicated a clear persistence of growth following 24-hour incubation in the wound dressing, indicating reduced time dependent antimicrobial efficacy after this period. In vitro studies [5] had shown silver ion concentration in simulated wound fluid to stay at biocidal levels for up to 7 days. This suggested that wound dressing time dependent antimicrobial efficacy was not reduced over the course of incubation. The observed persistence could be the result of adaptation to biocidal conditions present in wound dressings —however, this requires further experimentation. Additionally, persistence could indicate opportunistic growth under favorable growth conditions, such as reduced bacterial population density or increased oxygen. The adaptive culture [30] for both *S. aureus* and *P. aeruginosa* were well documented in the literature. These organisms were shown to respond to both antibiotic and biocidal metal exposure [31] by shifting phenotypes to extend survivability. Regarding *S. aureus*, for example, multidrug efflux pumps (MEPs) are shown to be overexpressed in response to biocidal metal exposure. The pumps served to expel ionic metals and ultimately increased bacterial persistence. The opportunistic growth of *S. aureus* was also underscored [32] by its asymptomatic colonization of roughly 30% of adults, despite an estimated 54% prevalence in skin and soft tissue infections (SSIs).

More generally, a viable-but-non-culturable (VBNCs) phenotype [19] could be related to the persistence of these bacteria in metal ion containing wound dressings. VBNC [22] was previously shown to be an adaptive response to biocide exposure, such as antibiotic exposure, and it has been shown to increase bacterial persistence under these conditions. Thus, phenotypic changes, including VBNC organisms may account for viable organisms at 3 and 7 days. This could explain the difference in quantitative results between qRT-PCR and cell culture assays.

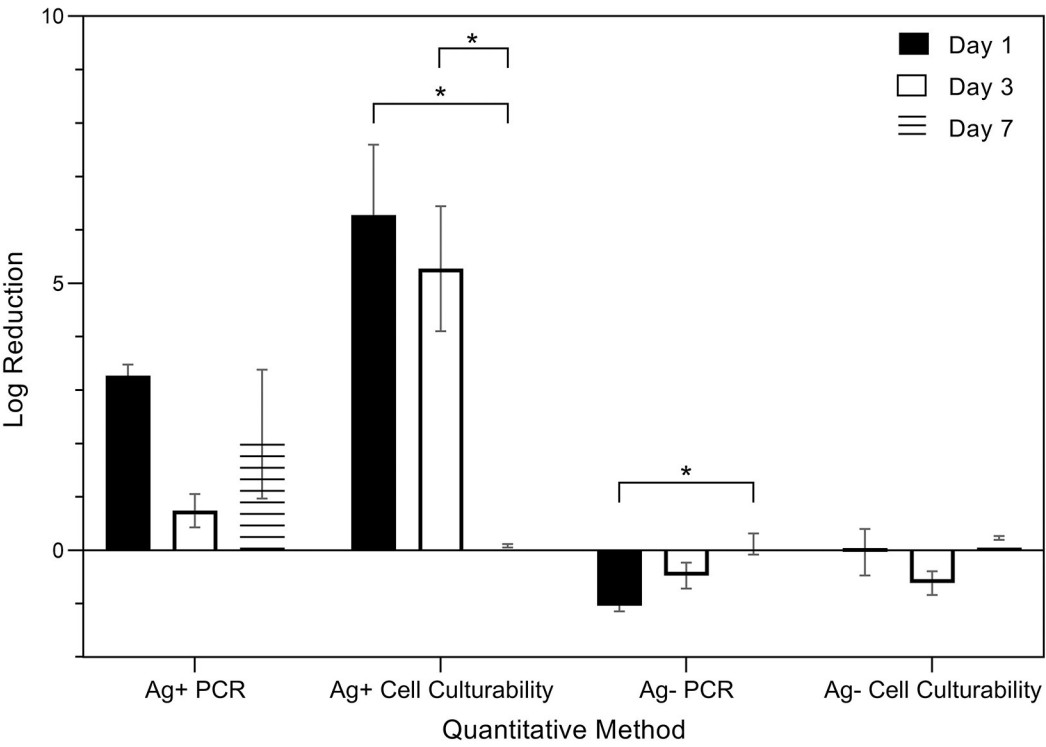

**Fig 9. Log reduction of P. aeruginosa by day.**

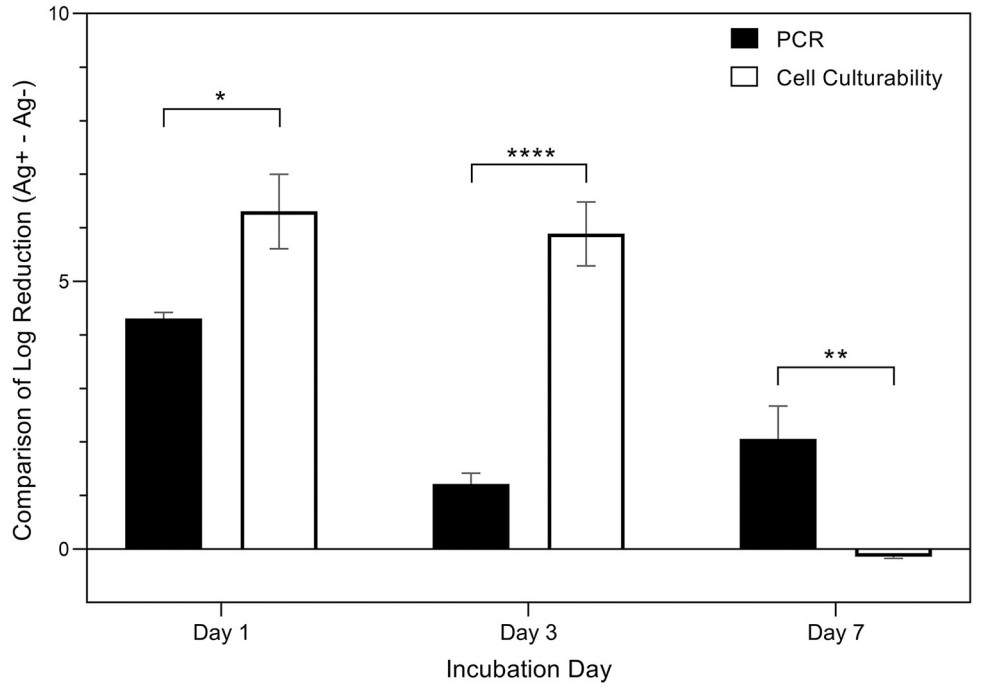

**Fig 10. Comparison of log reduction of P. aeruginosa by quantitative method.**

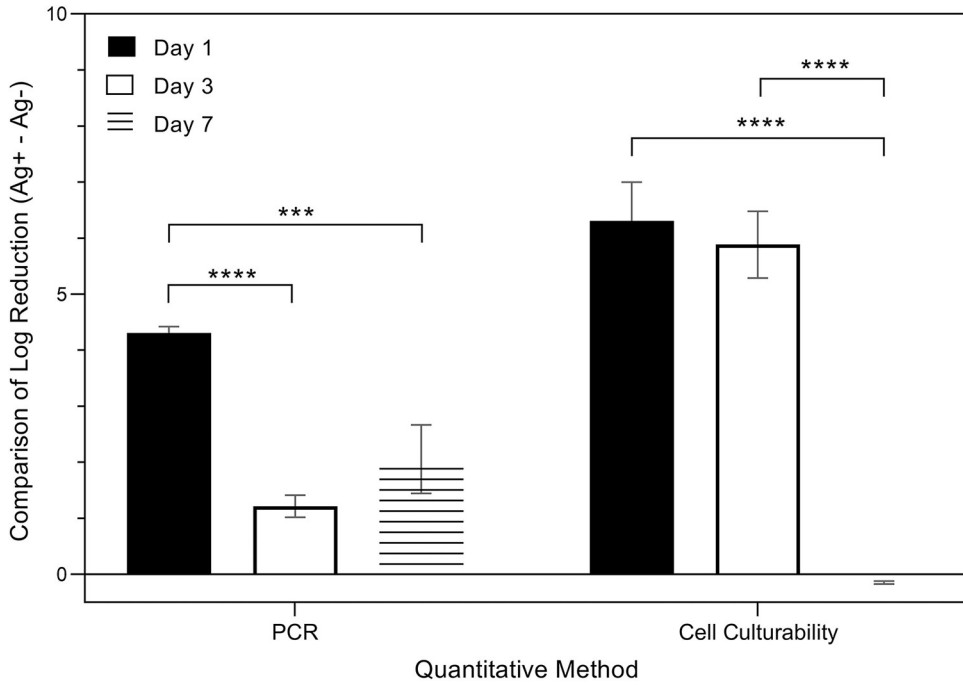

**Fig 11. Comparison of log reduction of P. aeruginosa by day.**

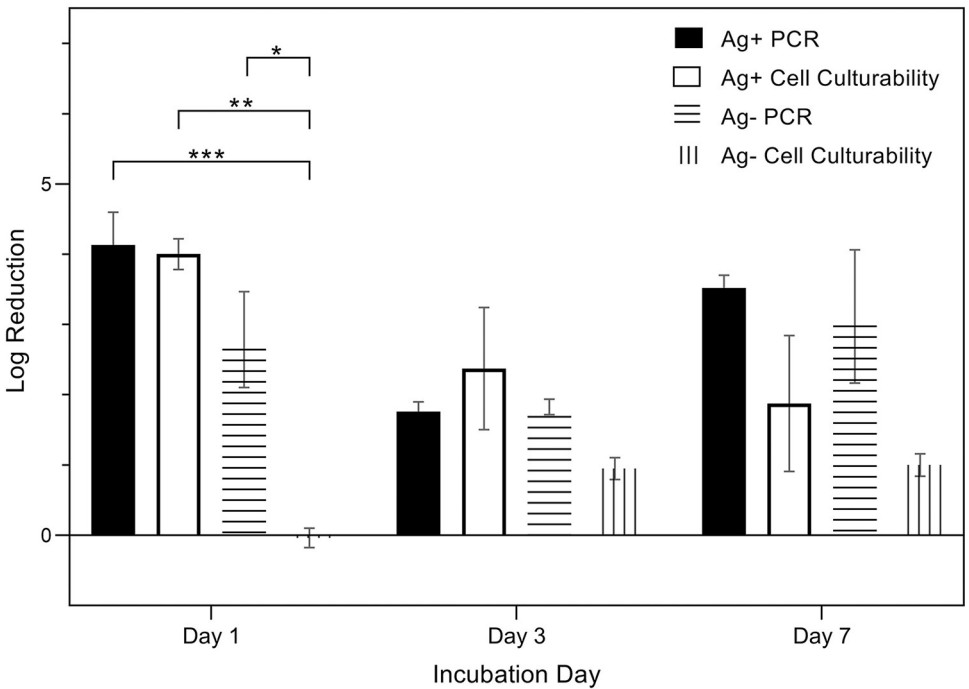

**Fig 12. Log reduction of S. aureus by quantitative method.**

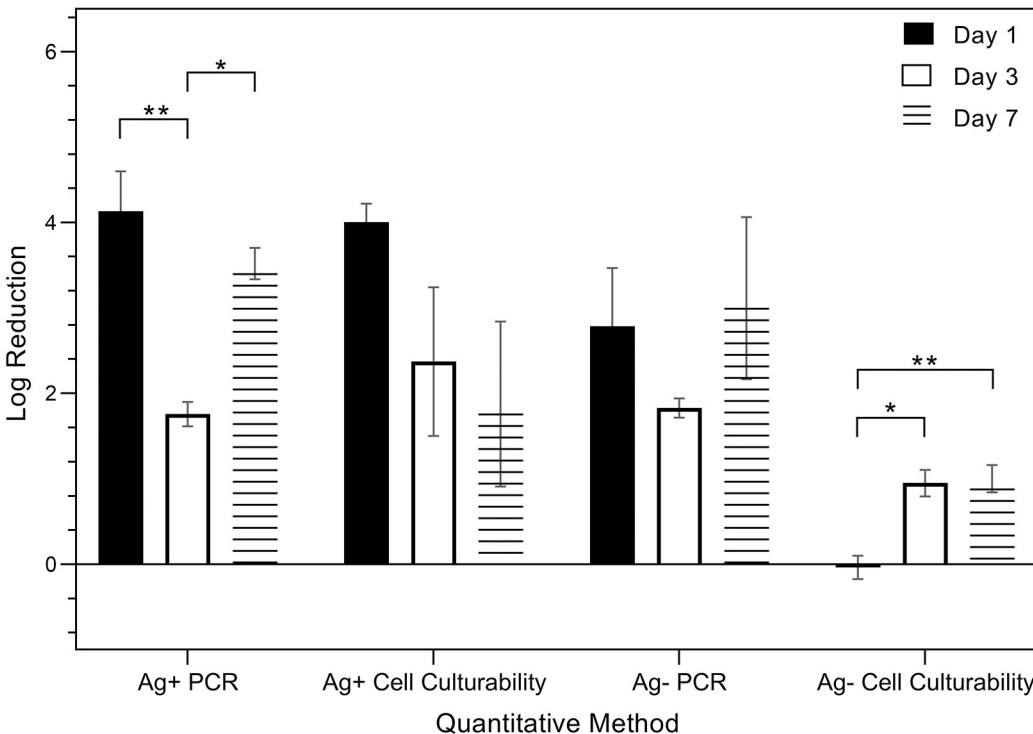

**Fig 13. Log reduction of S. aureus by day.**

However, without assays specific to VBNC biomarkers, the presence or absence of VBNC bacteria would require further investigation.

qRT-PCR results of both log and comparison of log reduction generally had smaller mean and standard deviation than those of cell culture assay, indicating that qRT-PCR could be a more conservative and reliable method. Furthermore, cell culture measured log reduction were consistently higher, suggesting inherent differences in sensitivities to bacterial quantity between two methods. Particularly, cell culture methods could underestimate [23] bacterial quantity, and overestimate the biocidal efficacy of wound dressings. This could be due to difficulty with culturing VBNCs [18] and/or other bacteria with distinct persistence phenotypes. Additionally, qRT-PCR results generally displayed smaller standard deviation than those of cell culture assays, indicating lower statistical variation. For many of the replicates for cell culture, zero colony counts were observed, and therefore increasing the variation in the average log reduction. Conversely, qRT-PCR, detected cDNA (mRNA expression analysis) from viable cells and more conservative and consistent log reduction results were observed. Log reduction and comparison of log reduction showed that the deviation between the results for each method was most noticeable with testing at 24 hours. For *P. aeruginosa* and *S. aureus*, the difference in quantitative results between incubation days are not statistically significant. However, the comparison of log reductions of both *P. aeruginosa* and *S. aureus* were shown to be statistically significant based on ANOVA and Paired t-test (p<0.005).

## Conclusion

In conclusion, incorporating longer time points for testing time dependent antimicrobial efficacy, the use of molecular assays, including qRT-PCR to measure for bacterial viability, along

## Comparison of Log Reduction by Method (S. aureus)

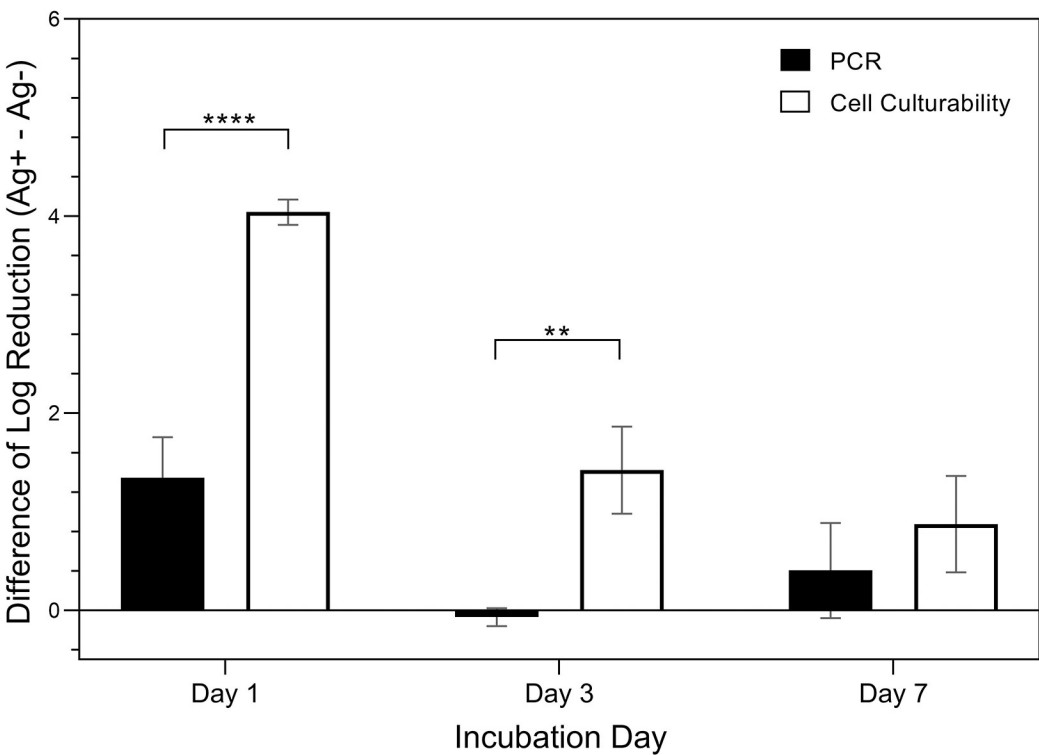

**Fig 14. Comparison of Log reduction of S. by quantitative method.**

## Comparison of Log Reduction by Day (S. aureus)

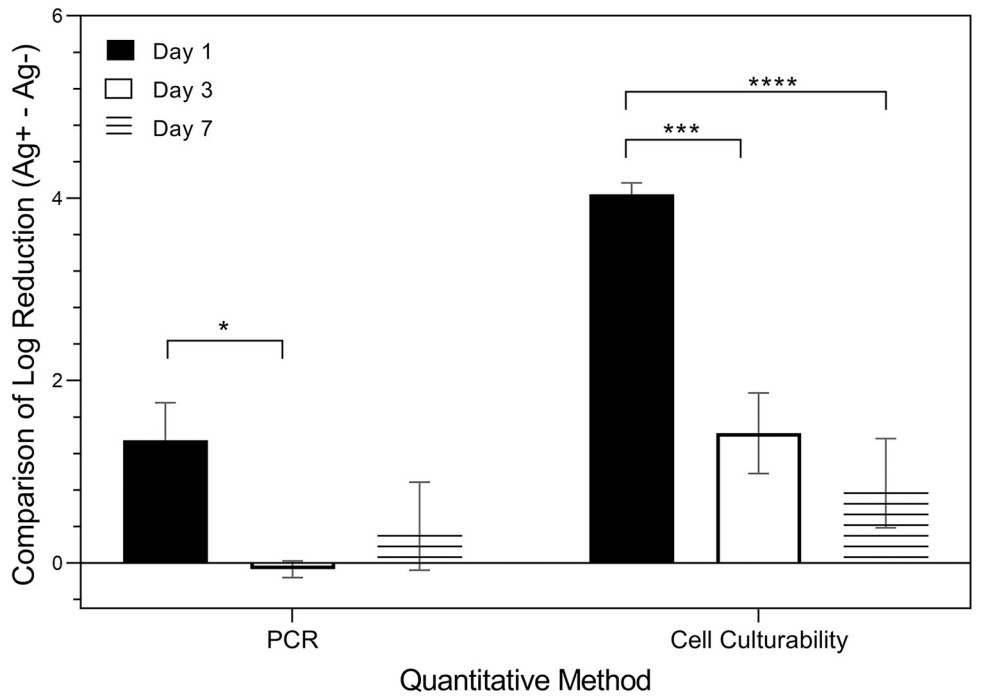

**Fig 15. Comparison of Log reduction of S. by day.**

with cell culture could better assist product testing for silver ion containing wound dressings. Hence, these methods in parallel with culturing could result in more reliable assessment of various emerging wound care products [26,29,33–37], ultimately reducing healthcare costs for patients and providers, and product efficacy. Application of PCR techniques such as relative quantification may further improve the proposed method.

## Supporting information

**S1 File. Minimal data set for absolute quantification.**
(XLSX)

## Acknowledgments

We appreciate insights provided in the development of this research by Drs. Anne Lucas, Steve Wood, and Michael Larranaga, Whiting School of Engineering at JHU, and Division of Biology, Chemistry and Material Science (DBCMS) at US Food and Drug Administration /Office of Science and Engineering Laboratories / Center for Device and Radiological Health.

### Disclaimer

The findings and conclusions in this paper have not been formally disseminated by the Food and Drug Administration and should not be construed to represent any agency determination or policy. The mention of commercial products, their sources, or their use in connection with material reported herein is not to be construed as either an actual or implied endorsement of such products by Department of Health and Human Services. The authors have declared that no competing interests exist.

## Author Contributions

**Conceptualization:** Enusha Karunasena.

**Data curation:** Sang Hyuk Lee, Thomas Glover, Nathan Lavey, Xiao Fu, Marc Donohue, Enusha Karunasena.

**Formal analysis:** Sang Hyuk Lee, Thomas Glover, Xiao Fu.

**Funding acquisition:** Enusha Karunasena.

**Investigation:** Sang Hyuk Lee, Thomas Glover.

**Methodology:** Sang Hyuk Lee, Thomas Glover, Nathan Lavey, Marc Donohue, Enusha Karunasena.

**Project administration:** Nathan Lavey, Enusha Karunasena.

**Resources:** Nathan Lavey, Enusha Karunasena.

**Software:** Sang Hyuk Lee, Xiao Fu.

**Supervision:** Nathan Lavey, Marc Donohue, Enusha Karunasena.

**Validation:** Sang Hyuk Lee, Thomas Glover, Nathan Lavey, Xiao Fu, Marc Donohue, Enusha Karunasena.

**Visualization:** Sang Hyuk Lee, Thomas Glover, Xiao Fu.

**Writing – original draft:** Sang Hyuk Lee, Thomas Glover.

**Writing – review & editing:** Sang Hyuk Lee, Marc Donohue, Enusha Karunasena.

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
