## [Decision Letter · Decision Letter 0]

1 Oct 2023

PONE-D-23-17876Application of modified AATCC-100 on silver ion containing wound dressing evaluation: Pseudomonas aeruginosa and Staphylococcus aureus modelsPLOS ONE 

Dear Dr. Lee,

Thank you for submitting your manuscript to PLOS ONE. After careful consideration, we feel that it has merit but does not fully meet PLOS ONE’s publication criteria as it currently stands. Therefore, we invite you to submit a revised version of the manuscript that addresses the points raised during the review process.

In light of the reviews there are issues that need to be addressed, including considerable work on the way the data is presented. More experiments are required as instructed by reviewer #2. All changes by both reviewer #1 and #2 are required.

The manuscript is of interest, so if the authors can address the reviewers' recommendations including the suggested additional experiments, it might be considered for publication.

We look forward to receiving your revised manuscript.

Kind regards,

Nourtan F. Abdeltawab, Ph.D.

Academic Editor

PLOS ONE

Journal Requirements:

 "I acknowledge, with profound gratitude, essential support from Whiting School of Engineering at JHU, the Critical Path 2020, McMi Funds, and Division of Biology, Chemistry and Material Science at US FDA CDRH."  

3. Please expand the acronym “FDA” (as indicated in your financial disclosure) so that it states the name of your funders in full.

"I acknowledge, with profound gratitude, essential support from Whiting School of Engineering at JHU, the Critical Path 2020, McMi Funds, and Division of Biology, Chemistry and Material Science at US FDA CDRH. Also, I appreciate keen insights from Dr. Anne Lucas, Dr. Steve Wood, and Dr. Michael Larranaga."

"I acknowledge, with profound gratitude, essential support from Whiting School of Engineering at JHU, the Critical Path 2020, McMi Funds, and Division of Biology, Chemistry and Material Science at US FDA CDRH."

5. We note you have included a table to which you do not refer in the text of your manuscript. Please ensure that you refer to Table 1 in your text; if accepted, production will need this reference to link the reader to the Table.

Reviewers' comments:

Reviewer's Responses to Questions

**Comments to the Author**

1. Is the manuscript technically sound, and do the data support the conclusions?

Reviewer #1: Yes

Reviewer #2: Partly

2. Has the statistical analysis been performed appropriately and rigorously? 

Reviewer #1: Yes

Reviewer #2: No

3. Have the authors made all data underlying the findings in their manuscript fully available?

Reviewer #1: Yes

Reviewer #2: Yes

4. Is the manuscript presented in an intelligible fashion and written in standard English?

Reviewer #1: No

Reviewer #2: Yes

5. Review Comments to the Author

Reviewer #1: the introduction section needs to be summarized and rearranged again

the methods section you didn't mention any reference even many of the methods are already known . Beside that you don't need to mention the full method in details if it is known just mention the reference.

the figures should be provided in more clear form as the resolution is very bad.

the Supplemental information section: the data presented in this section is more than the data you presented in the original paper, so please add these data to the original results not as a supplement

the statistical data and figure you illustrate in the supplementary information should be translated and represented in the paper result and discussion not as a Supplemental information.

Reviewer #2: 1- General comment: This is an in-vitro method to evaluate wound dressing. But this work should have included an in-vivo animal model to actually evaluate the wound dressing in presence of biological fluids such as pus (which results from bacterial infections). In addition, the absorption of silver ion through the skin is another factor that wasn’t taken into consideration. That’s why I strongly suggest performing an in-vivo animal model with wound infection.

2- The whole document should be represented as numbered lines to facilitate the reviewing process.

3- Specific comments:

Title It should include the word “in-vitro” to be more precise.

Abstract Is very vague and subjective without mentioning any of the results. I suggest adding major findings (with numbers) in addition to a conclusion sentence.

Introduction It should be written as a continuous section without sub-headings. The paragraphs must show a reasonable flow as you present the whole subject. Each paragraph represents an idea “but without a heading”. By the end of the intro the reader should know the aim of the study.

Materials and methods • Please number each heading and sub-headings with multilevel list headings.

• Please show the positive and negative controls for each experiment and explain the statistical evaluation for each exp.

• I suggest that you perform another experiment showing the release of silver ion from the wound dressing in-vitro over time and in simulated in-vivo experiments.

• Also, the concentration of silver ion in the skin and spleen of the animal in the in-vivo animal model should be measured to evaluate the actual release.

Results • As it was mentioned earlier, the culture technique was evaluated as “log reduction”, so I suggest that the growth curve for both S. aureus and P. aeruginosa represented as “Log CFU/ml vs time” instead of “CFU/mL”.

• Figure S7: the data should have been analyzed by either:

One-way ANOVA (if you compare each factor in a single day) followed by post test and multiple comparisons or TWO-way ANOVA (if you compare each factor among the three days) followed by post test for corrections and multiple comparisons.

• All the figures showing more than 2 factors should have been analyzed by one-way ANOVA followed by a post test for corrections and multiple comparisons.

Discussion • The first paragraph has no references, so please add recent references (2022 or 2023) to any sentence that is not your actual findings.

• In other paragraphs please add recent references and mention recent studies.

Conclusion I suggest you add a couple of sentences indicating your future directions.

References Please update the references list with more recent papers (2022 or 2023).

6. PLOS authors have the option to publish the peer review history of their article (what does this mean?). If published, this will include your full peer review and any attached files.

Reviewer #1: No

Reviewer #2: No

---

## [Author Response · Author response to Decision Letter 0]

9 Nov 2023

Hello. I appreciate the constructive comments. Please find the revised manuscript and relevant document in this resubmission. Please let me know if there is anything I can further clarify about our work. Thank you for your consideration!

1. Fig 1 is mentioned in text.

---

## [Editor Report · Decision Letter 1]

12 Nov 2023

PONE-D-23-17876R1Modified in-vitro AATCC-100 procedure to measure viable bacteria from wound dressingsPLOS ONE

Dear Dr. Lee,

Thank you for submitting your revised manuscript to PLOS ONE.

We invite you to submit a revised version of the manuscript that addresses the points raised during the review process.

You must include a point-by-point response to each of the reviewer's comments before we evaluate your revisions.

In addition, any changes you made to your revised manuscript must be highlighted using "track changes".

Your manuscript will not be handled further until above mentioned requirements are submitted.

You must follow instructions of this email carefully to avoid further delays in evaluation of your manuscript.

We look forward to receiving your revised manuscript.

Kind regards,

Nourtan F. Abdeltawab, Ph.D.

Academic Editor

PLOS ONE

---

## [Author Response · Author response to Decision Letter 1]

29 Dec 2023

Dear PLOS ONE Editor,

I appreciate the comment and patience. 

This resubmission packet contains requested documents: 

point to point response to reviewers,

Manuscript, 

Manuscript with tacked changes.

Please let me know if there is anything else I can clarify about our work. 

Thank you for your care.

Happy Holidays and New Year!

Sincerely, 

Sang Hyuk Lee PhD

---

## [Decision Letter · Decision Letter 2]

31 Jan 2024

Modified in-vitro AATCC-100 procedure to measure viable bacteria from wound dressings

PONE-D-23-17876R2

Dear Dr. Lee,

We’re pleased to inform you that your manuscript has been judged scientifically suitable for publication and will be formally accepted for publication once it meets all outstanding technical requirements.

Kind regards,

Nourtan F. Abdeltawab, Ph.D.

Academic Editor

PLOS ONE

Additional Editor Comments (optional):

Reviewers' comments:

Reviewer's Responses to Questions

**Comments to the Author**

1. If the authors have adequately addressed your comments raised in a previous round of review and you feel that this manuscript is now acceptable for publication, you may indicate that here to bypass the “Comments to the Author” section, enter your conflict of interest statement in the “Confidential to Editor” section, and submit your "Accept" recommendation.

Reviewer #3: All comments have been addressed

2. Is the manuscript technically sound, and do the data support the conclusions?

Reviewer #3: Yes

3. Has the statistical analysis been performed appropriately and rigorously? 

Reviewer #3: Yes

4. Have the authors made all data underlying the findings in their manuscript fully available?

Reviewer #3: Yes

5. Is the manuscript presented in an intelligible fashion and written in standard English?

Reviewer #3: Yes

6. Review Comments to the Author

Reviewer #3: (No Response)

7. PLOS authors have the option to publish the peer review history of their article (what does this mean?). If published, this will include your full peer review and any attached files.

Reviewer #3: No

---

## [Editor Report · Acceptance letter]

12 Mar 2024

PONE-D-23-17876R2 

PLOS ONE

Dear Dr. Lee, 

I'm pleased to inform you that your manuscript has been deemed suitable for publication in PLOS ONE. Congratulations! Your manuscript is now being handed over to our production team.

Kind regards, 

on behalf of

Dr. Nourtan F. Abdeltawab 

Academic Editor

PLOS ONE